# Crk proteins transduce FGF signaling to promote lens fiber cell elongation

Tamica N Collins[1†‡], Yingyu Mao[1†], Hongge Li[1§], Michael Bouaziz[1], Angela Hong[1], Gen-Sheng Feng[2], Fen Wang[3], Lawrence A Quilliam[4], Lin Chen[5], Taeju Park[6], Tom Curran[6], Xin Zhang[1]*

[1]Departments of Ophthalmology, Pathology and Cell Biology, Columbia University, New York, United States; [2]Department of Pathology, University of California San Diego, La Jolla, United States; [3]Center for Cancer Biology and Nutrition, Houston, United States; [4]Department of Biochemistry and Molecular Biology, Indiana University School of Medicine, Indianapolis, United States; [5]Department of Rehabilitation Medicine, Third Military Medical University, Chongqing, China; [6]The Children's Research Institute, Children's Mercy Kansas City, Kansas City, United States

*For correspondence:
xz2369@columbia.edu

[†]These authors contributed equally to this work

Present address: [‡]Department of Molecular Genetics and Cell Biology, University of Chicago, Chicago, United States; [§]Department of Microbiology and Immunology, Indiana University School of Medicine, Indianapolis, United States

Competing interests: The authors declare that no competing interests exist.

**Abstract** Specific cell shapes are fundamental to the organization and function of multicellular organisms. Fibroblast Growth Factor (FGF) signaling induces the elongation of lens fiber cells during vertebrate lens development. Nonetheless, exactly how this extracellular FGF signal is transmitted to the cytoskeletal network has previously not been determined. Here, we show that the Crk family of adaptor proteins, Crk and CrkI, are required for mouse lens morphogenesis but not differentiation. Genetic ablation and epistasis experiments demonstrated that *Crk* and *CrkI* play overlapping roles downstream of FGF signaling in order to regulate lens fiber cell elongation. Upon FGF stimulation, Crk proteins were found to interact with Frs2, Shp2 and Grb2. The loss of Crk proteins was partially compensated for by the activation of Ras and Rac signaling. These results reveal that Crk proteins are important partners of the Frs2/Shp2/Grb2 complex in mediating FGF signaling, specifically promoting cell shape changes.
DOI: https://doi.org/10.7554/eLife.32586.001

## Introduction

During the development of complex multicellular organisms, changes in epithelial cell morphology are essential for the tissue-specific cell differentiation patterning that leads to the subsequent formation of functional organs (*Settleman and Baum, 2008*). This is particularly clear when considering the formation of the ocular lens, which has served as a model system to delineate many developmental pathways (*Cvekl and Zhang, 2017*; *Gunhaga, 2011*). The development of the mouse lens begins at embryonic day 9.5 when the optic vesicle extends toward the presumptive lens ectoderm, inducing the latter to thicken into a cuboidal layer of epithelial cells commonly referred to as the lens placode. At E10.5, the cells making up the lens placode undergo apical constriction to form the lens pit, which eventually closes at its anterior surface to form the lens vesicle (*Chauhan et al., 2011*). After the newly differentiated primary fiber cells extend anteriorly from the posterior region of the lens vesicle, the anterior epithelial cells begin to migrate posteriorly towards the equatorial region of the lens, at which point they begin to differentiate into secondary fiber cells that continue to occupy the space within the lens interior. This process is accompanied by up to a 1000-fold increase in the length of the newly formed secondary fiber cells, which coordinate with the primary fiber cells to organize an elegant concave pattern that maintains the structural integrity and transparency of the mature lens (*Bassnett, 2005*; *McAvoy et al., 1999*; *Sue Menko, 2002*).

**eLife digest** As an embryo develops, its cells divide multiple times to transform into the specialized cell types that form our tissues and organs. To carry out specific roles, cells need to be of a certain shape. For example, in mammals, the cells that make up the main portion of the eye lens, develop into a fiber-like shape to be perfectly aligned with each other. This enables them to transmit light to the retina at the rear end of the eye. To do so, the lens cells increase over 1000 times in length with the help of a group of proteins called the Fibroblast Growth Factor, or FGF for short.

The FGF pathway includes a network of interacting proteins that transmit signals to molecules inside the lens cells to control how they specialize and grow. However, until now it was not clear how it does this. Here, Zhang et al. used mouse lens-cells grown in the laboratory to investigate how FGF signaling causes cells to change their structure. The experiments revealed two related proteins called Crk and Crkl that linked the FGF pathway with another signaling system. When these two proteins were removed from the lens cells, the lens cells were still able to specialize, but could no longer grow in length. This suggests that these two processes are independent of each other.

Moreover, Crk and Crkl helped the cells to change shape by increasing the amount of another group of proteins called Ras, which are known to both help cells to specialize and to regulate their shape. Zhang et al. discovered that the amount of Ras proteins determined whether cells specialized or modified their shape by changing the organization of proteins in the cell.

Millions of children are born with cataracts, a disease caused when lens cells fail to shape properly. A better knowledge of FGF signaling may help to understand how cataracts develop and inspire future treatments. Moreover, the pathways identified in this study could also apply to other organs and diseases in which FGF signaling is active.

DOI: https://doi.org/10.7554/eLife.32586.002

Previously proposed models that have been considered for the mechanism by which lens fiber cells elongate consist of microtubule reorganization, increased cell volume and actin dynamics (*Audette et al., 2017*). Microtubules are prominent components of the cytoskeleton lining the plasma membrane of lens fiber cells. They are oriented longitudinally along the lens with their minus end towards the anterior pole and their plus ends facing the posterior (*Byers and Porter, 1964*). which is consistent with the presence of microtubule organizing centers at the apical ends of lens fiber cells (*Lo et al., 2003*). However, Beebe et al reported that Nocodazole inhibition of microtubule polymerization failed to disrupt fiber cell elongation in chick lens explants (*Beebe et al., 1979*). Instead, they proposed that the expansion of cell volume was the key mechanism by which fiber cells elongated to form the functional mature lens. However, utilizing precise measurement techniques, it was later found that there was no apparent increase in lens volume in vivo at the onset of fiber cell elongation (*Bassnett, 2005*). The differentiation of lens epithelial cells into lens fiber cells is also associated with the assembly of actin filaments beneath the cortical membrane (*Weber and Menko, 2006*). When this cortical actin structure was inhibited by cytochalasin D in lens explants, both the differentiation and elongation of lens cells were blocked. Nevertheless, neither the disruption of the cell adhesion molecules N-cadherin or β1-integrin nor the ablation of the actin regulators Rac1 and Rho completely prevented the lengthening of lens fiber cells (*Logan et al., 2017*; *Maddala et al., 2011*, *2015*; *Pathania et al., 2016*; *Pontoriero et al., 2009*). Consequently, how the actin cytoskeleton is controlled during fiber cell elongation has remained an open question.

Fibroblast Growth Factor (FGF) signaling is known to play an important role in lens cell differentiation and elongation. Transgenic expression of human FGF-1 using an αA-crystallin promoter was found to induce lens epithelial cells to acquire elongated shapes and fiber cell characteristics (*Robinson et al., 1995*). In explant cultures, FGF was also found to promote lens fiber cell differentiation and elongation in a dose dependent manner (*Lovicu and McAvoy, 2001*; *McAvoy and Chamberlain, 1989*). Conversely, genetic ablation of FGF receptors leads to a complete loss of lens cell differentiation and elongation (*Zhao et al., 2008*). Several proteins have been implicated in the direct engagement with these active FGF receptors, including Frs2/3, Grb14, Shb, PLCγ and Crk (*Brewer et al., 2015*; *Klint and Claesson-Welsh, 1999*). Of particular interests are Crk and the

related protein Crkl, which are mammalian homologs of the viral *Crk* oncogene that prossess the ability to promote the tyrosine phosphorylation of cellular proteins (*Feller, 2001*). Lacking intrinsic tyrosine kinase activity, the Crk family of proteins act as adaptors that transduce signals from upstream phosphotyrosine-containing proteins to downstream SH3-interacting partners (*Birge et al., 2009*). Biochemical studies have shown that FGF2-stimulated endothelial cell proliferation is dependent on the binding of Crk to the phosphorylated tyrosine residue 463 in FGFR1 (*Larsson et al., 1999*). In line with this finding, *Crk* null mice display some of the cardiovascular and cranial features of Noonan syndrome, which is caused by aberrant Ras-MAPK signaling (*Park et al., 2006*; *Roberts et al., 2007*; *Schubbert et al., 2006*; *Tartaglia et al., 2001*; *Tartaglia et al., 2007*). Crkl was also identified as a component of an FGF8-induced feed forward loop, resulting in anchorage-independent cell growth (*Seo et al., 2009*). Consistent with this, the human *CRKL* gene lies within the chromosome 22q11 deletion region that causes DiGeorge syndrome, which shares the pharyngeal and cardiac defects seen in *Fgf8*-deficient mice (*Moon et al., 2006*). Despite these findings implicating Crk and Crkl in FGF signaling, a recent study has shown that mutating their putative Y463 binding site in Fgfr1 did not produce any observable phenotype in transgenic mice

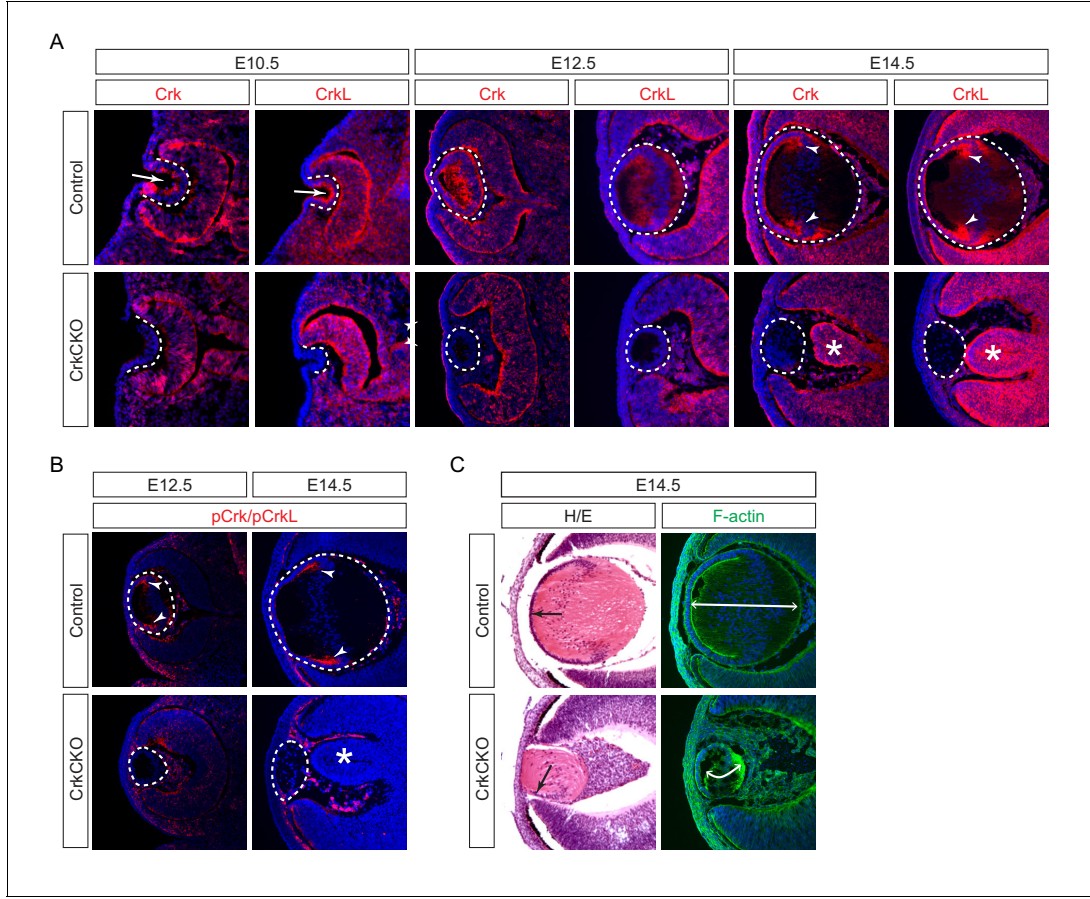

**Figure 1.** Crk and Crkl are essential for lens development. (**A**) Crk and Crkl immunostaining were localized to the invaginating lens vesicle at E10.5 (arrows) and to the elongating lens fiber cells near the transitional zone of the lens at E14.5 (arrowheads). These staining patterns were specifically lost in the CrkCKO lens. The dotted lines enclose the region of the lens and the disorganization of the retina was marked with asterisks (**B**) The phosphorylation of both Crk and Crkl was noticeably absent in the CrkCKO lens (arrowheads). (**C**) The CrkCKO lens size was significantly reduced with the anterior lens epithelium rotated sideways (arrows) and the disorganized lens fiber cells markedly shortened (double headed arrows).

DOI: https://doi.org/10.7554/eLife.32586.003

The following figure supplement is available for figure 1:

**Figure supplement 1.** *Crk* and *Crkl* single mutants did not display lens phenotype.
DOI: https://doi.org/10.7554/eLife.32586.004

(*Brewer et al., 2015*). Therefore, the potential role and mechanism of Crk proteins in FGF signaling remain uncertain.

In this study, we showed that the lens specific knockout of *Crk* and *Crkl* disrupted lens fiber cell elongation without affecting differentiation, suggesting that lens cell morphogenesis can be uncoupled from differentiation during development. FGF loss- and gain-of-function experiments demonstrated that Crk proteins act downstream of FGF signaling to enhance ERK phosphorylation. Contrary to the previous belief that Crk proteins directly bind to the Fgfr, we found that mutating the purported Crk docking site on Fgfr1 failed to perturb lens development or Crk phosphorylation. Instead, our data showed that Crkl was recruited to the Frs2/Shp2/Grb2 complex after FGF stimulation. Crk/Crkl deficient animals phenocopied Rac1 but not Rap1 mutants, and activation of Rac1 and Ras signaling partially reversed the observed lens elongation defects caused by the deletion of Crk and Crkl. These results show that the Crk family of adaptor proteins are essential partners of the Frs2/Shp2/Grb2 complex that forms during FGF signaling, and are specifically required for stimulating the actin reorganization that is necessary for the morphological shaping of lens cells.

## Results

### Ablation of Crk and Crkl caused lens defects

We observed that Crk and Crkl proteins displayed a restricted localization pattern in the lens. At E10.5, Crk and Crkl were predominantly confined to the apical side of the lens vesicle (*Figure 1A*, arrows), away from the basal side where integrins interact with the basement membrane (*Figure 1A*, dotted lines). By contrast, Crk and Crkl exhibited a more diffuse pattern at E12.5 when the posterior lens vesicle cells gave rise to the primary lens fibers (*Figure 1A*). However, by E14.5, Crk and Crkl were specifically enriched in the transitional zone where the lens epithelial cells begin to differentiate and elongate into the secondary lens fiber cells (*Figure 1A*, arrowheads). Using an antibody that recognizes the phosphorylated forms of both of these proteins, we were able to observe that the phosphorylation of Crk and Crkl also mainly occurs in the transition zone of the lens at this stage of development (*Figure 1B*, arrowheads). These results suggest that Crk activity is under dynamic regulation as the lens cells undergo successive morphological changes during development.

We next ablated *Crk* genes using *Pax6^Le^-Cre*, also known as *Le-Cre,* which is initially active in the lens placode and later in the lens epithelium (*Ashery-Padan et al., 2000*). As expected, this resulted in the loss of both Crk/Crkl and pCrk/pCrkl in the *Pax6^Le^-Cre;Crk^flox/flox^;Crkl^flox/flox^* (CrkCKO) lens after E10.5 (*Figure 1A and B*, dotted line). Although deletion of either *Crk* or *Crkl* alone did not perturb lens development (*Figure 1—figure supplement 1A–C*), the CrkCKO lens displayed a reduction in lens size, rotation of the lens epithelial layer, and disorganization of the lens fiber cells at E14.5 (*Figure 1C*, arrow). Using F-actin staining to better delineate individual cell shapes, we observed a significant reduction in the length of the lens fiber cells (*Figure 1C*). In addition to these fully penetrant lens phenotypes, the neural retina was often observed to aberrantly protrude toward the diminished lens. This observation is consistent with the known role of the properly developed lens in the correct placement of the retina (*Figure 1A*, asterisks) (*Ashery-Padan et al., 2000*).

During lens development, the transcription factors Pax6 and Prox1 are of vital importance for the normal differentiation of the transparent lens. Pax6 controls lens induction and fate determination while Prox1 regulates fiber cell differentiation and crystallin expression (*Ashery-Padan et al., 2000*; *Audette et al., 2016*). Interestingly, notwithstanding the severe morphological defects including the frequent ventral rotation of the lens epithelial layer, Prox1, Pax6 and multiple forms of crystallin (α,β,γ) were still expressed (*Figure 2A and B*). In addition, the polarity of the lens fiber cells was also preserved as evidenced by the localization of Zo-1 and β1 integrin to the apical and basal sides of the lens, respectively (*Figure 2C*, arrowheads). However, there were both a reduction in cell proliferation and an increase in apoptosis in the lens epithelial cells as shown by Ki67 and TUNEL staining (*Figure 2D and E*, arrowheads), which likely accounted for the diminished lens epithelial layer (*Figure 2A and B*, arrowheads). Collectively, these data show that Crk proteins are dispensable for lens differentiation and polarity, but are essential for proliferation, survival and elongation of the cells that make up the structurally mature lens.

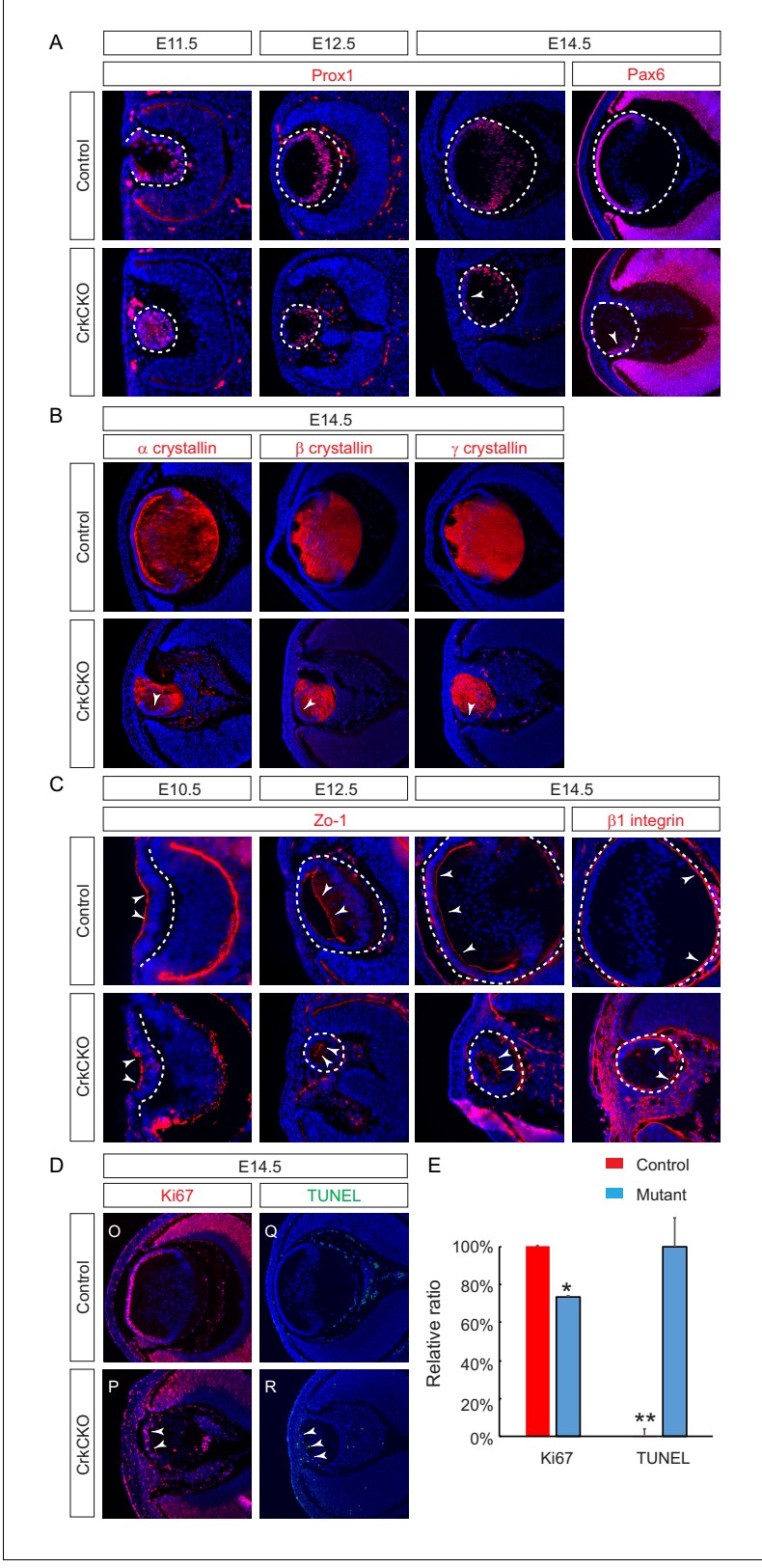

**Figure 2.** Molecular defects in the *Crk* and *Crkl* double mutant lens. (**A**) There were no significant changes in the staining intensity of the lens determinant markers Prox1 and Pax6. (**B**) Further, none of the three forms of Crystallins (α, β, γ) displayed any changes in staining intensity in the CrkCKO lens. (**C**) The polarity of the CrkCKO lens fiber cells was maintained as indicated by both the apical expression of Zo-1 and the basal expression of β1

*Figure 2 continued on next page*

*Figure 2 continued*

integrin. (D) The number of Ki67-expressing proliferative cells was significantly decreased and the number of TUNEL-positive apoptotic cells was increased (arrowheads). (E) Quantification of proliferation and apoptosis in wild type and CrkCKO lens. Student's t test, *p<0.01, **p<0.001, n = 4.

DOI: https://doi.org/10.7554/eLife.32586.005

The following source data is available for figure 2:

**Source data 1.** Source data for *Figure 2E*.

DOI: https://doi.org/10.7554/eLife.32586.006

## Crk proteins act downstream of FGF signaling to control fiber cell elongation

To investigate the role of Crk proteins in FGF signaling, we took a three-prong approach, combining in vitro, in vivo and ex vivo experiments. First, we performed loss-of-function experiments to examine whether FGF signaling is required for Crk protein activity. Consistent with previous studies

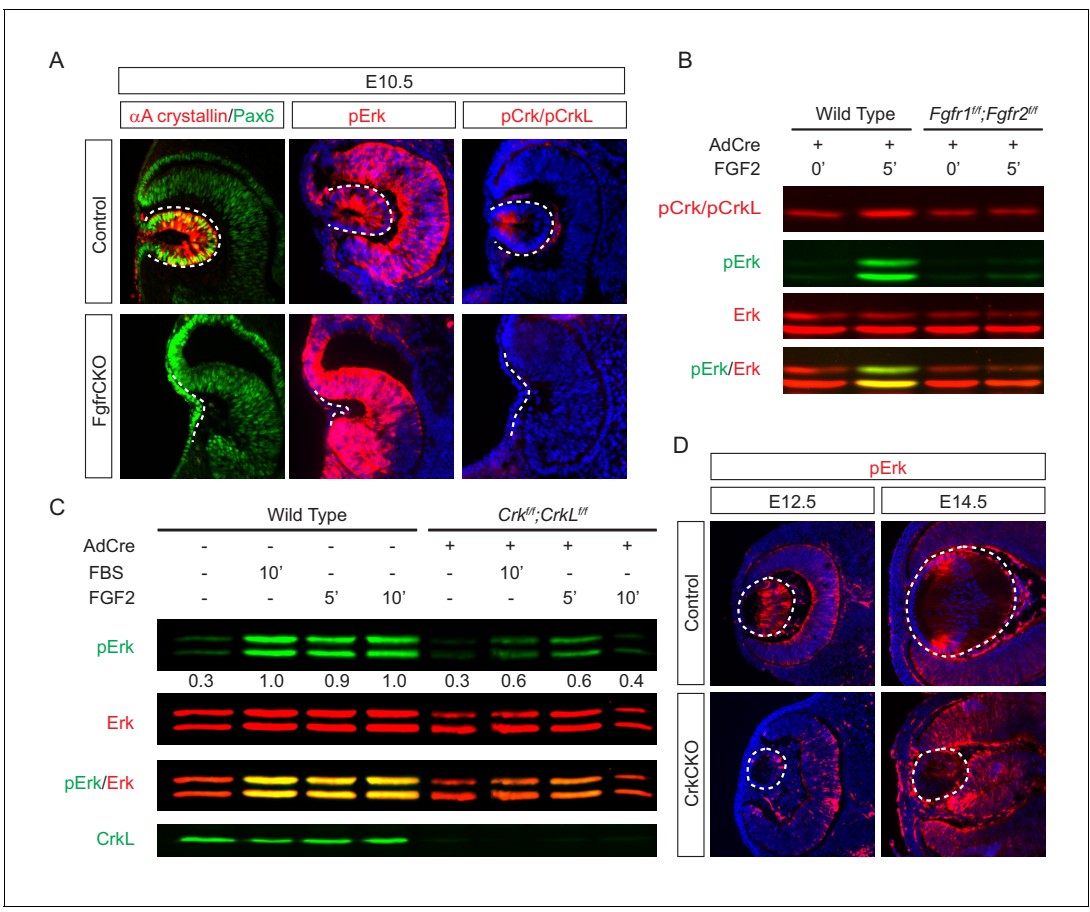

**Figure 3.** Crk proteins mediate FGF signaling in Erk phosphorylation. (A) Genetic ablation of Fgfr1 and Fgfr2 disrupted the proper formation of the lens vesicle with the phosphorylation of Erk and Crk/Crkl proteins being noticeably absent. (B) Mouse Embryonic fibroblast (MEF) cells treated with FGF2 displayed an increase in pCrk/Crkl and pErk levels, which were abrogated by the removal of Fgfr1 and Fgfr2 using a Cre-expressing adenovirus. (C) Ablation of Crk proteins in MEF cells reduced FGF2-induced Erk phosphorylation. The pErk/Erk ratios were noted below the pERK blot. (D) The CrkCKO lens displayed a significant decrease in pERK staining compared to the wild type lens.

DOI: https://doi.org/10.7554/eLife.32586.007

The following figure supplement is available for figure 3:

**Figure supplement 1.** The Y463F mutation in *Fgfr1* (*Fgfr1$^{Crk}$*) did not affect the phosphorylation of Crk and Erk proteins that is essential for lens development.

DOI: https://doi.org/10.7554/eLife.32586.008

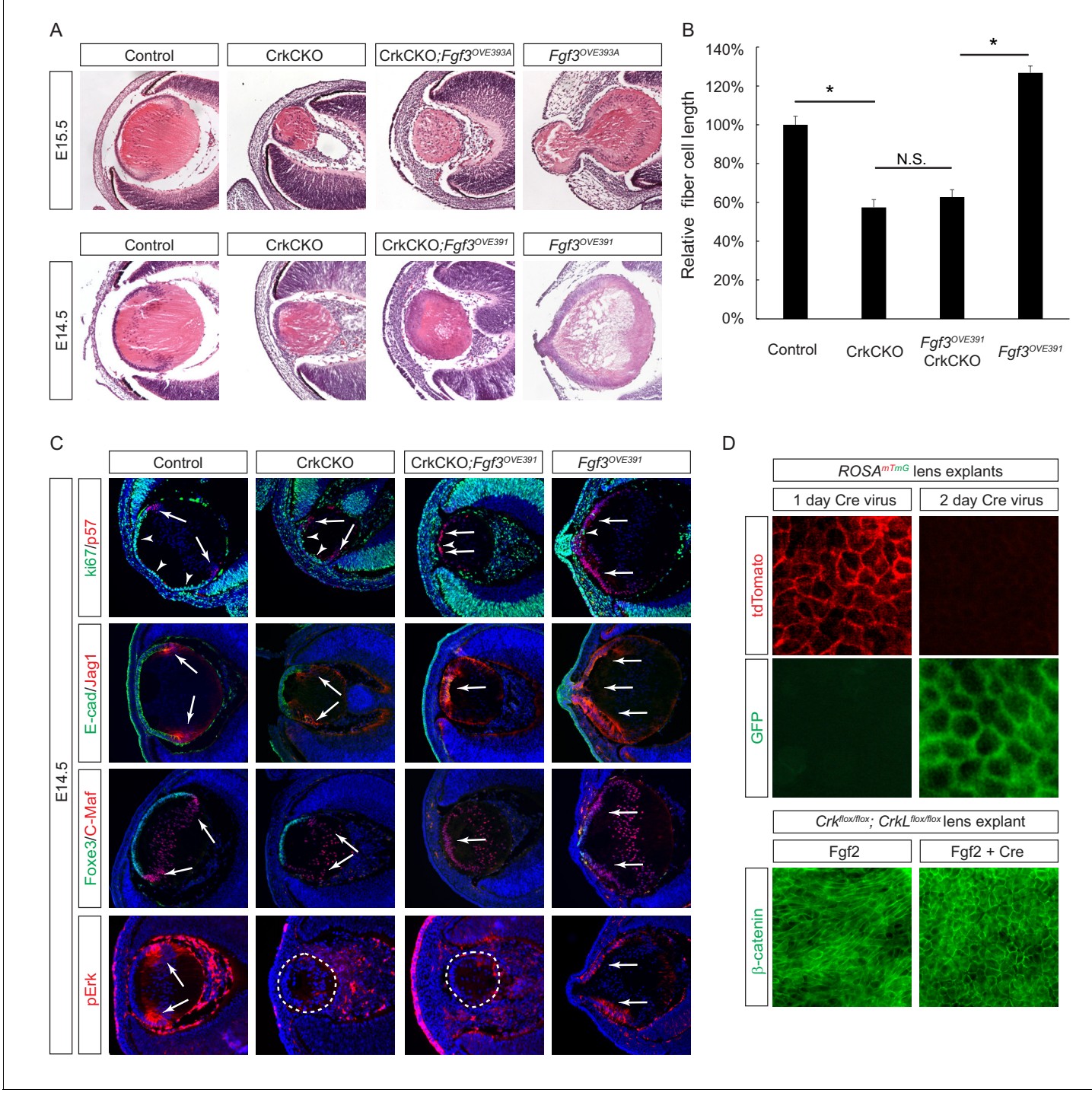

**Figure 4.** Crk/Crkl deletion prevented FGF-induced lens cell elongation without affecting differentiation. (**A**) Overexpression of *Fgf3* in *Fgf3^OVE391^* and *Fgf3^OVE393A^* strains resulted in increased lens fiber cell elongation and overall lens sizes. However, both of these phenotypes were suppressed after being crossed with the CrkCKO mutant, indicating a genetic epistasis interaction between the FGF and Crk signaling pathways. (**B**) Quantification of lens fiber cell length. One-way ANOVA test followed by Tukey's multiple comparisons test, *p<0.01, *n* = 3. (**C**) Deletion of the *Crk* genes did not prevent the premature differentiation phenotype observed in the *Fgf3* overexpressing lens, as indicated by a reduction of lens progenitor cell markers (Ki67, E-cad and Foxe3) and an increase of differentiation cell markers (p57, Jag1, and C-Maf) within the presumptive lens epithelial layer (arrows). Nonetheless, pERK staining was significantly reduced in the CrkCKO;*Fgf3^OVE391^* lens. (**D**) The Cre-expressing adenovirus induces efficient genetic recombination in lens explant cultures as indicated by the *ROSA^mTmG^* reporter. In *Crk^flox/flox^;Crkl^floxflox^* explants, the Cre-mediated deletion of Crk proteins prevented Fgf2 from inducing cell shape changes.

*Figure 4 continued on next page*

*Figure 4 continued*

DOI: https://doi.org/10.7554/eLife.32586.009

The following source data is available for figure 4:

**Source data 1.** Source data for *Figure 4B* and *Figure 7P*.

DOI: https://doi.org/10.7554/eLife.32586.010

showing that FGF signaling is essential for early lens development (*Garcia et al., 2011*), we observed that genetic ablation of Fgfr1/2 in *Pax6^{Le}-Cre;Fgfr1^{flox/flox};Fgfr2^{floxflox}* (FgfrCKO) mutants disrupted lens vesicle invagination at E10.5 (*Figure 3A*). Although the mutant lens cells still expressed Pax6, the lens specific marker αA-crystallin was not induced. Importantly, whereas the control embryos displayed the phosphorylation of Erk and Crk in the invaginating lens vesicle, these staining patterns were absent in FgfrCKO lens cells (*Figure 3A*). To further corroborate these results in vitro, we isolated primary mouse embryonic fibroblast (MEF) cells from *Fgfr1^{flox/flox};Fgfr2^{floxflox}* embryos and infected them with a Cre-expressing adenovirus to ablate Fgfr1 and Fgfr2. Unlike wild type controls, these mutant MEF cells failed to elevate the level of pCrk/Crkl and pErk upon FGF stimulation, demonstrating that both Crk and Erk proteins are under the tight regulation of FGF signaling (*Figure 3B*). To probe the relationship between Crk and Erk, we next infected *Crk^{flox/flox};Crkl^{floxflox}* MEF cells with the Cre adenovirus to deplete Crk proteins (*Figure 3C*). As a result, both the intensity and duration of FGF-stimulated Erk phosphorylation were down regulated, suggesting that Crk proteins modify the quantity of FGF-ERK signaling by elevating and prolonging ERK activation. Consistent with this, we noticed that Erk phosphorylation was prominent in the elongating primary lens fiber cells at E12.5 and in the transitional zone of the lens at E14.5 (*Figure 3D*). In the CrkCKO mutant lens, however, pERK staining was significantly reduced (*Figure 3D*). Together, this data shows that Crk proteins regulate FGF-induced Erk activation in the developing lens.

The second approach we took to probe the role of *Crk* genes in FGF signaling was based on gain-of-function experiments. We utilized two *Fgf3* transgenes that are driven by the αA-crystallin promoter to target the lens (*Robinson et al., 1998*), which led to an anterior expansion of pErk staining (*Figure 4C*). The *Fgf3^{OVE393A}* line displayed excessive elongation of the lens fiber cells that protruded through the corneal epithelium, while the *Fgf3^{OVE391}* line showed a more modest enlargement of the lens (*Figure 4A*). After crossing these mice with *Crk* mutants, however, lens abnormalities in both lines were suppressed and ectopic phospho-Erk staining was abrogated. The length of the fiber cells was reduced to the same size as those seen in the CrkCKO mutants, demonstrating that *Crk* genes were necessary for the induction of fiber cell elongation by FGF (*Figure 4B*). *Fgf3* overexpression also led to premature exiting of the cell cycle as indicated by the loss of the cell proliferation marker Ki67 (*Figure 4C*, arrowheads) and an increase in expression of the cell cycle inhibitor p57 (*Figure 4C*, arrows). As a result, the anterior lens epithelial cells differentiated prematurely to express the fiber cell markers Jag1 and C-Maf at the expense of the epithelial cell markers E-cadherin and Foxe3. Interestingly, the cell cycle and differentiation abnormalities were not rescued after crossing the *Fgf3* transgenic mice with CrkCKO mutants. These genetic epistasis experiments further highlighted that the specific functionality of *Crk* genes is in mediating FGF signaling for lens cell elongation but not differentiation.

In the third approach, we used the mouse lens explant system to test the direct role of Crk proteins in FGF-induced fiber cell elongation (*Korol et al., 2014*). In this assay, the explant lens isolated from P3 mouse embryos carrying *Crk^{flox/flox};Crkl^{floxflox}* alleles was infected with a Cre-expressing adenovirus to achieve acute genetic ablation. This avoids any potential complication that may stem from defects in early lens development or other compartments of the eye. Using a *ROSA^{mTmG}* mouse line that switches the reporter expression from membrane TdTomato to membrane GFP upon Cre mediated recombination, we showed that a 2 day incubation period with the virus was sufficient to induce genetic changes in all lens epithelial cells (*Figure 4D*). In control explants, β-catenin staining revealed a robust elongation of the lens epithelial cells after FGF2 exposure. By contrast, the lens epithelial cells in *Crk^{flox/flox};Crkl^{floxflox}* explants treated with the Cre adenovirus retained the epithelial specific hexagonal shape without any obvious signs of elongation. Taken together, these three lines of evidence established that Crk proteins are essential mediators of FGF signaling whose specific function relates to the fiber cell elongation that occurs during lens development.

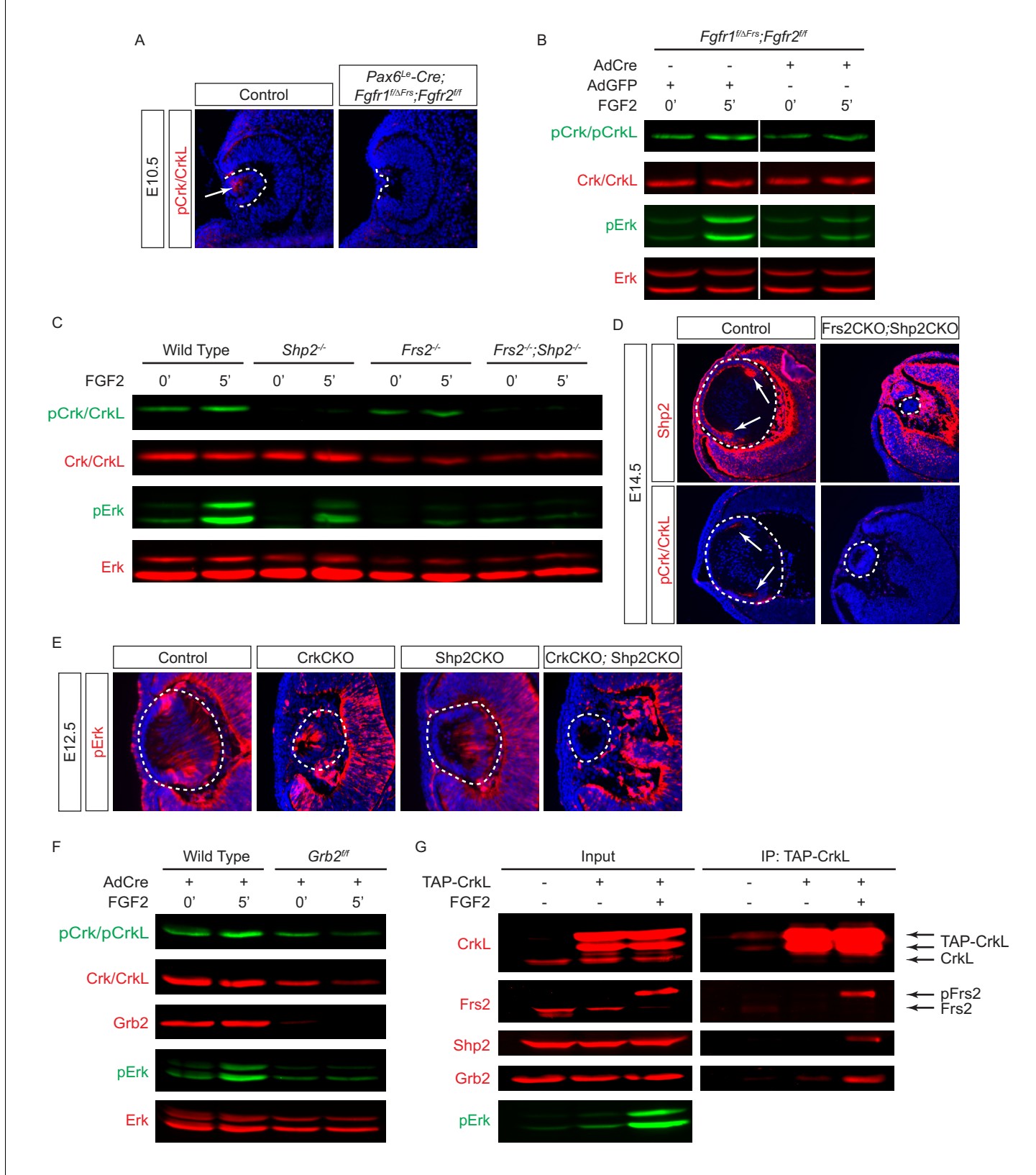

**Figure 5.** Crk proteins are recruited to the Frs2-Shp2-Grb2 complex in FGF signaling. (A–B) Mutating the Frs2-bindng site in *Fgfr1$^{ΔFrs}$* resulted in the loss of pCrk/Crkl in the *Pax6$^{Le}$-Cre;Fgfr1$^{flox/ΔFrs}$;Fgfr2$^{floxflox}$* mutant lens. (B) FGF2 was unable to induce the phosphorylation of Crk proteins in *Fgfr1$^{flox/Δ}$$^{Frs}$;Fgfr2$^{floxflox}$* MEF cells after treatment with the Cre expressing adenovirus. (C) FGF2-induced pCrk/Crkl and pErk were significantly downregulated in

*Figure 5 continued on next page*

Figure 5 continued

both *Frs2* and *Shp2* null MEF cells. (D) Shp2 was successfully depleted in the Frs2CKO;Shp2CKO lens, which resulted in the loss of pCrk/Crkl staining. (E) pERK was downregulated in both CrkCKO and Shp2CKO lenses and was further reduced in CrkCKO;Shp2CKO mutants. (F) FGF2-induced pCrk/Crkl and pErk were down regulated in *Grb2* deficient MEF cells. (G) TAP-taged Crkl pulled down Frs2, Shp2 and Grb2 after FGF2 stimulation. Note that only the slower moving phosphorylated form of Frs2 successfully interacted with Crkl.

DOI: https://doi.org/10.7554/eLife.32586.011

## The Frs2 binding site on Fgfr1 is required for Crk signaling

Previous studies have suggested that Crk proteins bind directly to the phosphorylated tyrosine-463 residue on Fgfr1 to mediate downstream signaling (*Larsson et al., 1999*). Surprisingly, mice carrying a Y463F mutation in *Fgfr1* (*Fgfr1$^{Crk}$*) lacked any observable phenotypes and were reported to be both viable and fertile (*Brewer et al., 2015*). The lack of abnormality in *Fgfr1$^{CRK}$* mice prompted us to examine whether it was due to the compensatory functions of other Fgf receptors. We isolated MEF cells from *Fgfr1$^{Crk}$;Fgfr2$^{flox}$* mice and removed *Fgfr2* by Cre-mediated recombination in vitro. Although we have shown above that genetic inactivation of Fgfr1 and Fgfr2 together was sufficient to abrogate FGF signaling in MEF cells (*Figure 3B*), FGF was still able to induce phosphorylation of Crk/Crkl and Erk in *Fgfr1$^{CRK}$* MEF cells after the deletion of *Fgfr2* (*Figure 3—figure supplement 1A*). We next investigated the requirement of the Y463 residue of Fgfr1 for Crk signaling in lens development. For this purpose, we genetically ablated all FGF receptors (*Fgfr2*, *Fgfr3* and *Fgfr4*) with the exception of *Fgfr1* in the mouse lens, resulting in only a modest reduction in lens size (*Figure 3—figure supplement 1B*). Even in such a stringent genetic background, a homozygous *Fgfr1$^{Crk}$* mutation did not further worsen the lens phenotype or disrupt pCrk staining. These results show that the putative Y463 Crk binding site on Fgfr1 is dispensable for FGF-dependent lens development and Crk signaling.

Since our biochemical and genetic data did not support a functional role for the direct interaction between FGF receptors and the Crk protein, we next explored whether FGF signaling may engage Crk indirectly through intermediaries. Frs2 is a myristylated protein located at the plasma membrane that binds to Fgf receptors specifically at a juxtamembrane site. With multiple tyrosine residues phosphorylated by activated Fgf receptors, Frs2 acts as a nexus of FGF signaling by presenting easily accessible docking sites for the phosphotyrosine-binding proteins Shp2 and Grb2, which in turn activate Ras-MAPK signaling. Taking advantage of a mutant *Fgfr1* allele (*Fgfr1$^{\Delta Frs}$*) that lacks the Frs2 binding site (*Hoch and Soriano, 2006*), we showed that formation of the lens vesicle was indeed disrupted in *Pax6$^{Le}$-Cre;Fgfr1$^{flox/\Delta Frs}$;Fgfr2$^{flox/flox}$* embryos (*Figure 5A*), which resembled the phenotype of FgfrCKO null mutants (*Figure 3A*). Importantly, we observed that the loss of the Fgfr-Frs2 interaction also abrogated the phosphorylation of Crk proteins. This was further confirmed in vitro using *Fgfr1$^{flox/\Delta Frs}$;Fgfr2$^{floxflox}$* MEF cells infected with a Cre adenovirus. The resulting *Fgfr1$^{-/\Delta Frs}$;Fgfr2$^{-/-}$* MEF cells failed to elevate the levels of pERK or pCrk/Crkl in response to FGF2 stimulation (*Figure 5B*), demonstrating that the Frs2-binding site on Fgfr1 is necessary for Crk signaling.

## Crk proteins are recruited indirectly to FGF receptors by the Frs2-Shp2-Grb2 complex

The above results suggested that Frs2 may be the adaptor protein that recruits Crk to Fgf receptors. To test this idea, we first examined Crk phosphorylation in MEF cells that were mutated by the Cre-mediated deletion of *Frs2* and *Shp2*. As shown in *Figure 5C*, phosphorylation of Crk proteins induced by FGF was significantly reduced in *Frs2* null MEF cells. Moreover, in MEF cells lacking the Frs2 interacting protein Shp2, even the normally seen basal levels of pCrk were lost. We have previously showed that both Frs2 and its binding partner Shp2 have relatively slow turnover rates in vivo (*Li et al., 2014*). As a result, the conditional inactivation of *Frs2* or *Shp2* alone exhibited only a modest lens phenotype. Combined deletion of *Frs2* and *Shp2* in *Le-Cre;Frs2$^{flox/floxs}$;Shp2$^{floxflox}$* (Frs2CKO; Shp2CKO) embryos, however, did block lens development at E14.5. At this stage, the control lens displayed a significant accumulation of Shp2 protein in the transitional zone of the lens, which is also the area of maximum pErk and pCrk/Crkl staining (*Figure 5D*, arrows). In contrast, neither Shp2 nor the pCrk/Crkl proteins were detectable in the Frs2CKO;Shp2CKO lens. Notably, an in vitro ablation of either Crk or Shp2 reduced the Erk phosphorylation that is normally induced by FGF (*Figure 3C*

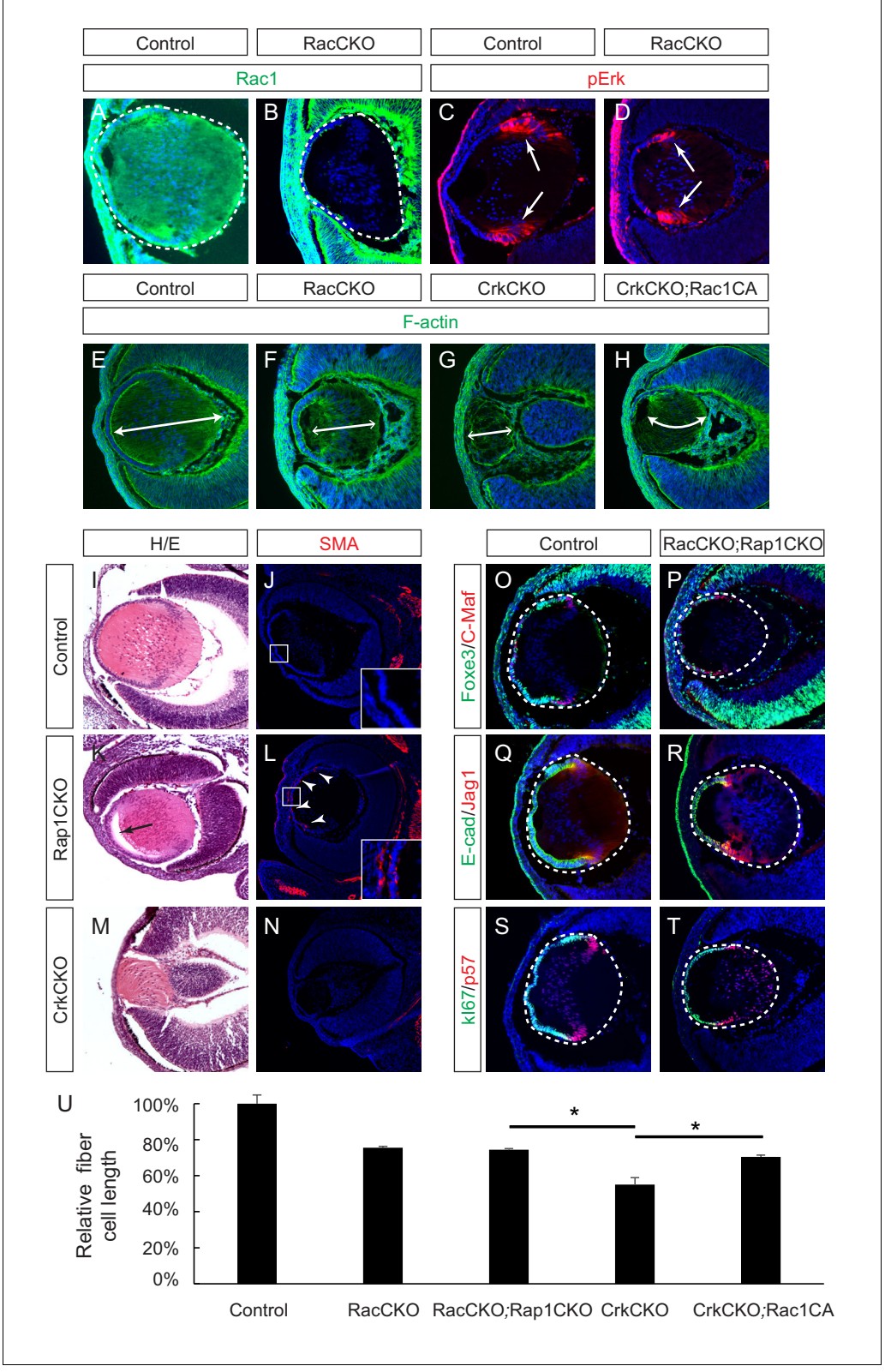

**Figure 6.** Rac proteins are downstream effectors of Crk signaling. (A–D) Staining of Rac-depleted lenses with a pErk antibody showed no significant difference in staining intensity. (E–H) Phalloidin (F-actin) staining revealed that the length of lens fiber cells (indicated by arrows) was significantly reduced in CrkCKO and RacCKO lenses, which was partially reversed after the activation of Rac signaling in CrkCKO;Rac1CA lenses. (I–N) The Rap1 depleted

*Figure 6 continued on next page*

*Figure 6 continued*

mutants displayed a detachment of the lens fiber cells from the anterior side of the lens epithelial cells (arrow) and Smooth Muscle Actin (SMA) staining within the lens epithelial layer itself (arrowheads). These phenotypes were absent both in the control and the Crk/Crkl depleted lenses. (O–T) Immunstaining of the Rac and Rap1 depleted lenses with progentior (Foxe3, E-cad, Ki67) and differentiation (C-Maf, Jag1,p57) markers did not reveal any defects in differentiation or any further shortening of the fiber cells. (I) Quantification of fiber cell lengths. One-way ANOVA test followed by Tukey's multiple comparisons test, *p<0.01, *n* = 3.

DOI: https://doi.org/10.7554/eLife.32586.012

The following source data is available for figure 6:

**Source data 1.** Source data for *Figure 6U*.

DOI: https://doi.org/10.7554/eLife.32586.013

and *Figure 5D*). Consistent with this, both CrkCKO and Shp2CKO embryos at E12.5 continued to display residual levels of pErk staining in the lens (*Figure 5E*). In CrkCKO;Shp2CKO mutants, however, pErk staining was entirely abolished, suggesting that Shp2 and the Crk family of proteins act synergistically to regulate Erk signaling.

Grb2 is another binding partner of Frs2 that is essential for the activation of Ras-MAPK signaling. In Grb2-depleted MEF cells, we observed a similar reduction in Crk and Erk signaling in response to FGF stimulation (*Figure 5F*). The attenuation of Crk phosphorylation in *Frs2*, *Shp2* and *Grb2* deficient cells raises the possibility that Crk proteins are recruited to the Fgf receptors via the Frs2-Shp2-Grb2 complex. To probe the physical interaction between these proteins, we transfected NIH-3T3 cells with a TAP-Crkl construct that encodes the Crkl protein conjugated to a tandem affinity purification (TAP) tag used for purification (*Hallock et al., 2015*). When the concentration of FGF was increased, we observed a shift of Frs2 mobility in our immunoblot analysis that is representative of the phosphorylated form of Frs2 being generated as has been previously reported (*Figure 5G*) (*Kouhara et al., 1997*). Consistent with this, there was also an increase in the levels of pErk present in the cell lysates. Interestingly, immunoprecipitated TAP-Crkl only pulled down the phosphorylated form of Frs2 induced by FGF stimulation, which was also accompanied by Shp2 and Grb2. Overall these results indicate that the Frs2-Shp2-Grb2 complex is responsible for recruiting Crk proteins to the activated Fgf receptor.

## Rac1 is a downstream effector of crk proteins in lens fiber cell elongation

Crk proteins have been implicated in the activation of the small molecular GTPases Rac1 and Rap1, which play important roles in cell adhesion, cytoskeletal rearrangement and cell shape changes (*Birge et al., 2009*; *Feller, 2001*). Previous studies have reported that the conditional deletion of Rac1 in the lens resulted in impaired actin polymerization that lead to a morphologically impaired lens (*Maddala et al., 2011*). Since *Rac2* has also been reported to be expressed in the lens (*Rao et al., 2004*), we decided to inactivate both *Rac1* and *Rac2* to compare their phonotype with that of the *Crk* mutants. Immunofluorescence confirmed the specific depletion of Rac1 in the $Pax6^{Le}$-$Cre;Rac1^{flox/flox};Rac2^{-/-}$ (RacCKO) lens (*Figure 6A–B*). Unlike the CrkCKO lens, however, the RacCKO lens did not display a decrease in pErk staining (*Figure 6C–D*). Nonetheless, depletion of Rac proteins resulted in a lens fiber cell elongation defect, albeit milder than that seen in the CrkCKO lens (*Figure 6E–G*). To further explore the relationship between Rac and Crk proteins, we utilized the $R26-Rac1^{LSL-G12V}$ allele that results in the expression of a constitutively active form of Rac1 (Rac1CA) after the Cre mediated excision of its STOP cassette (*Srinivasan et al., 2009*). In the $Pax6^{Le}$-$Cre;$ $Crk^{flox/flox};Crkl^{floxflox};R26-Rac1^{LSL-G12V}$ (CrkCKO;Rac1CA) lens, there was a statistically significant increase in fiber cell length as compared to that of the CrkCKO lens (*Figure 6H and U*). The partial rescue of the CrkCKO phenotype by Rac1 activation supports the idea that Rac1 is a downstream effector of Crk in the signaling pathways that lead to lens fiber cell elongation.

Rap1 is also known to be targeted by Crk proteins via activation of the guanine nucleotide exchange factor (GEF), C3G, and FGFR1 has previously been reported to activate Rap1 in endothelial cells (*Quilliam et al., 2002*; *Yan et al., 2008*). We reasoned that if Rap1 was a downstream effector of Crk proteins in FGF signaling, *Rap1* and *Crk* deficient lenses should phenocopy each

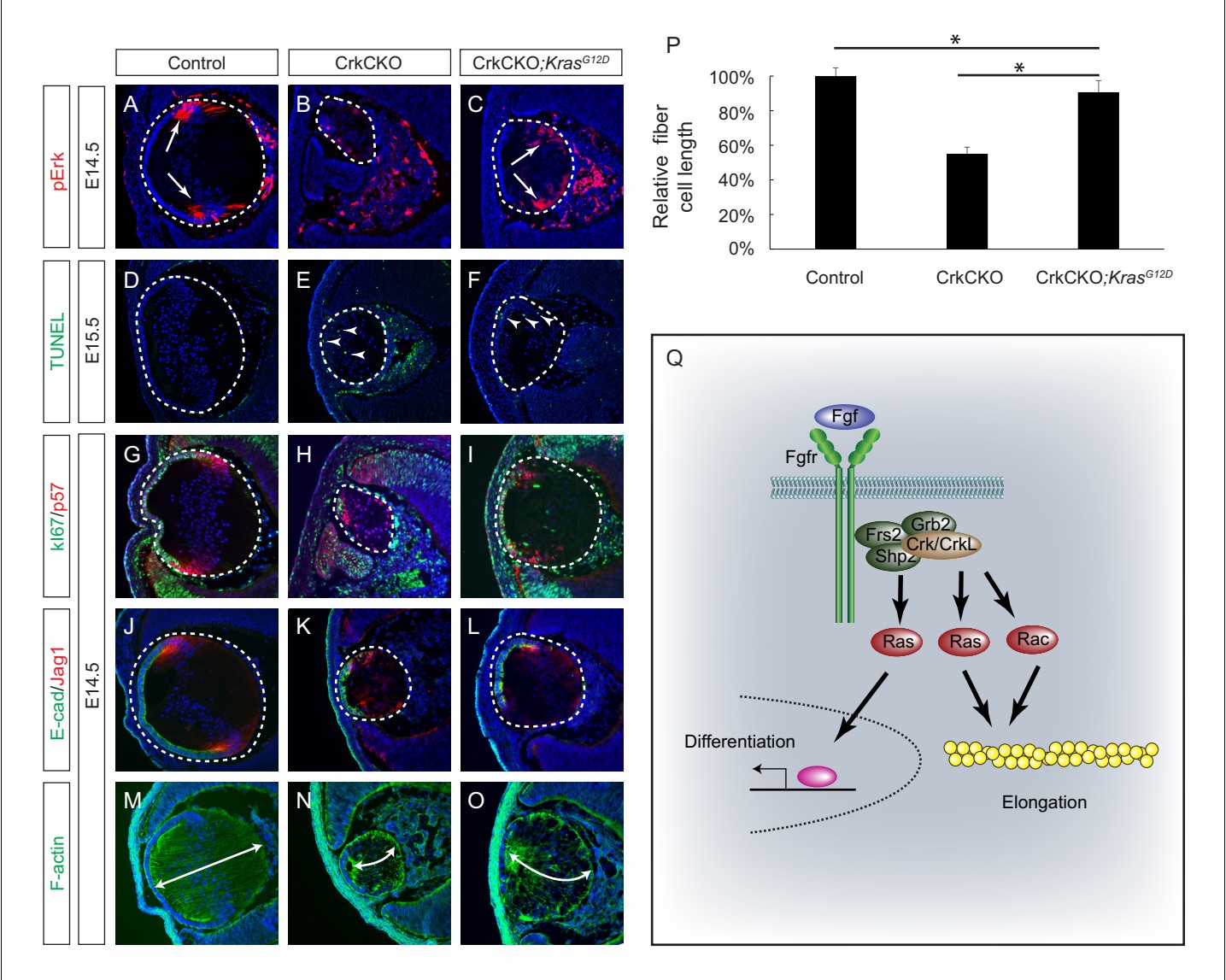

**Figure 7.** Constitutive Kras signaling can compensate for the loss of Crk and Crkl in lens development. (**A–C**) Despite the loss of Crk and Crkl, Erk phosphorylation was partially recovered in CrkCKO;Kras$^{G12D}$ lenses. (**D–F**) A signficant amount of TUNEL positive cells remained in both CrkCKO and CrkCKO;Kras$^{G12D}$ lenses. (**G–L**) Cell proliferation indicated by Ki57 increased in the CrkCKO;Kras$^{G12D}$ lens as compared to the CrkCKO lens, but there was no significant difference in staining intensity for the differentiation markers E-cad and Jag1. (**M–P**) Lens fiber cell length increased significantly in CrkCKO;Kras$^{G12D}$ lenses as compared to CrkCKO ones. Fiber cell length was measured based on F-actin staining and statistical analysis was performed using the one-way ANOVA test followed by Tukey's multiple comparisons test (*p<0.01, n = 3). (**Q**) Model of Crk function in FGF signaling. The binding of FGF to its receptor induces the assembly of the Frs2-Shp2-Grb2 complex, which subsequently activates Ras signaling to promote lens differentiation. When FGF signaling is further elevated at the transitional zone of the lens, Crk proteins were additionally recruited by the Frs2-Shp2-Grb2 complex to further promote Ras and Rac signaling, resulting in actin cytoskeletal rearrangement and cell shape changes.
DOI: https://doi.org/10.7554/eLife.32586.014

other as well. In agreement with a previous report (**Maddala et al., 2015**), we found that the conditional knockout of two *Rap1* genes (*Pax6*$^{Le}$-*Cre;Rap1a*$^{flox/flox}$;*Rap1b*$^{flox/flox}$ (Rap1CKO)) disrupted the epithelial polarity of the lens, leading to an ectopic expression of the epithelial-mesenchymal-transition (EMT) marker smooth muscle actin (SMA) in lens epithelium cells (**Figure 6I–L**). Due to cell adhesion defects, the posterior lens fiber cells also failed to remain attached to the anterior epithelial cells as is represented by a noticeable gap in the anterior part of the lens. In contrast, neither of these phenotypes were observed in the CrkCKO mutant lens (**Figure 6M and N**), which instead

displayed a more severe lens fiber cell elongation defect than the Rap1CKO mutants. We also considered the possibility that *Rap1* and *Rac* may be functionally redundant in the developing ocular lens. The combined deletion of these four genes in *Le-Cre;Rac1$^{flox/flox}$;Rac2$^{-/-}$;Rap1a$^{flox/flox}$;Rap1b$^{flox/flox}$* (RacCKO;Rap1CKO) embryos, however, did not produce any lens differentiation abnormalities or enhance the fiber cell elongation defects previously observed in the RacCKO mutants (*Figure 6O–U*). Taken together, these results argue against the Rap1 family of proteins being downstream targets of Crk signaling during lens development.

## Activation of ras signaling ameliorated lens fiber cell elongation defects present in the *Crk* mutant

The above experiments showed that a constitutively active form of Rac1 resulted in only modest attenuation of the lens fiber cell elongation defects seen in the CrkCKO mutant, suggesting that there exist additional downstream effectors of the Crk proteins that are important in regulating fiber cell shape. As we have observed a significant downregulation of pErk in the CrkCKO mutant lens, one potential candidate is the Ras-MAPK signaling pathway. We subsequently crossed the CrkCKO mutant with a *Kras$^{LSL-G12D}$* mouse, which harbors a Cre-inducible allele of the oncogenic G12D mutated *Kras* (*Tuveson et al., 2004*). Because this mutant is expressed from the endogenous *Kras* locus, it is expected to activate Ras signaling at a normal physiological level as opposed to being overexpressed. Consistent with this, we observed a modest increase in pERK staining in the *Pax6$^{Le}$-Cre;Crk$^{flox/flox}$;Crkl$^{flox/flox}$;Kras$^{LSL-G12D}$* (CrkCKO;Kras$^{G12D}$) lens (*Figure 7A–C*). Although TUNEL staining showed that the cell apoptosis defect was not rescued in the CrkCKO; Kras$^{G12D}$ lens, the number of proliferative Ki67 positive cells increased significantly, which can likely explain the extension of the lens epithelium as evidenced by E-cadherin staining (*Figure 7D–L*). Importantly, the length of the lens fiber cells in the CrkCKO;Kras$^{G12D}$ lens increased to about 90% of that of the control lens, demonstrating that the activation of Ras signaling can largely rescue the fiber elongation defects caused by the loss of Crk proteins (*Figure 7M–P*). This result provides strong genetic evidence that Ras signaling plays an essential role in Crk-mediated lens cell shape changes.

## Discussion

The ocular lens is derived from a single cell type that undergoes an orderly set of successive differentiation change that are represented by various cell shape alterations occurring along the developmental timeline. This makes it an excellent model for studying the cell signaling pathways involved in these morphological processes. Although FGF signaling is known to regulate both lens fiber cell differentiation and elongation, it is unclear whether these two functions can be separated at the mechanistic level. In this study, we successfully showed that the Crk family of adaptor proteins specifically mediate FGF signaling in the control of lens cell shape but not differentiation. Contrary to previous claims, we demonstrated that the putative Crk binding site on Fgfr1 is dispensable. Rather, the recruitment of Crk proteins to the Frs2-Shp2-Grb2 complex via the Frs2 docking site on Fgf receptors was observed. Our study further showed that the downstream Crk effectors involved in regulating the precise cytoarchitecture of the lens are primarily Ras and to a lesser extent Rac1, but not Rap1 (*Figure 7Q*). These results identify Crk proteins as the essential adapters that link FGF signaling to cytoskeletal dynamics.

This study also demonstrated that Crk and Crkl play essential overlapping functions in embryonic lens development. Whereas lack of either Crk or Crkl did not disrupt lens development in the mouse embryos, combined deletion of Crk and Crkl led to a profound defect in lens fiber cell elongation. There is a substantial overlap between Crk and Crkl in many biological functions including the Reelin pathway (*Park and Curran, 2008*), neuromuscular synapse formation (*Hallock et al., 2010*), migration of T cells to sites of inflammation (*Huang et al., 2015*) and podocyte morphogenesis (*George et al., 2014*). These studies and ours confirm that single and double floxed mice for Crk and Crkl are crucial models for investigating the distinct and overlapping biological functions of these closely related proteins.

Crk and Crkl are versatile adaptor proteins that can interact with a wide spectrum of signaling molecules (*Birge et al., 2009*). Previous studies have shown that they can be recruited by molecular scaffolds such as Dab1 to transmit Reelin signaling or p130Cas and paxillin to participate in integrin signaling (*Nojima et al., 1996*; *Petit et al., 2000*; *Sekine et al., 2012*). Following growth factor

stimulation, Crk and Crkl are also known to bind directly to several receptor tyrosine kinases (RTKs) via their SH2 domains and become rapidly phosphorylated at their specifically targeted tyrosine residues (Y221 in Crk and Y207 in Crkl) (*Antoku and Mayer, 2009*; *Feller et al., 1994*). Upon ligand stimulation, FGF receptors undergo autophosphorylation at multiple tyrosine residues, which serve as docking sites for downstream signaling proteins. Previous studies have suggested that Crk and Crkl can recognize a conserved phosphotyrosine residue in FGFRs (pY463 in FGFR1 and pY466 in FGFR2) through their SH2 domains (*Larsson et al., 1999*). By examining the Y463F mutant of Fgfr1 both in vitro and in vivo, we showed that this site was dispensable for FGF signaling with regards to the activation of Crk proteins. By contrast, we showed that Crk proteins interact with Frs2 and Shp2 proteins, and that their loss prevents FGF-induced Crk phosphorylation. This is reminiscent of Frs2's function in response to nerve growth factor (NGF) to assemble a complex containing Crk, C3G, Rap1 and Braf in order to prolong MAPK signaling (*Kao et al., 2001*). Interestingly, in that case, the binding of Crk requires the Shp2-docking site on Frs2. We have also identified Grb2 as yet another interacting partner of Crkl and showed that Grb2 was required for FGF-induced phosphorylation of Crk and Crkl. Since both Frs2 and Shp2 interact with Grb2, we would like to suggest that the assembly of the entire Frs2-Shp2-Grb2 complex is necessary for the recruitment of the Crk family of adaptor proteins to Fgf receptors.

C3G, Sos and Dock1 were the first three GEFs identified to be directly associated with the SH3 domains of Crk (*Hasegawa et al., 1996*; *Oda et al., 1994*; *Tanaka et al., 1994*). While Sos is primarily involved in the activation of Ras-MAPK signaling, C3G and Dock1 promote the exchange of GDP for GTP in the small GTPases Rap1 and Rac1 (*Birge et al., 2009*). These active GTP-bound proteins subsequently regulate integrin signaling and actin polymerization (*Gloerich and Bos, 2011*; *Ridley, 2011*). We showed that the *Crk* knockout did not recapitulate the *Rap1* mutant lens phenotype, ruling out Rap1 as a critical downstream effector of Crk proteins. Instead, activation of Rac1 and, more importantly, Ras effectively rescued the lens fiber elongation defect seen in *Crk* mutants. In addition to these results further clarifying the downstream targets of Crk in lens development, they also raise an interesting question regarding the nature of Ras signaling in this process. Previous studies have shown that the Frs2/Shp2 mediated Ras-MAPK pathway acts downstream of FGF signaling to regulate lens fiber cell differentiation (*Li et al., 2014*; *Madakashira et al., 2012*; *Upadhya et al., 2013*), but in this study, our genetic evidence demonstrated that the Crk-mediated Ras signaling pathway only promoted fiber cell elongation and not differentiation. The uncoupling of lens fiber cell differentiation and elongation have been previously observed in lens explants, where pharmacological inhibition of ERK suppressed the morphological changes induced by FGF signaling while not preventing the expression of differentiation markers such as β-crystallin (*Lovicu and McAvoy, 2001*). We propose that these results reveal a biphasic function of Ras signaling, which promotes either differentiation or elongation of lens cells in a dosage dependent manner. In this model, Ras signaling induced directly by the Frs2/Shp2/Grb2 complex is sufficient to stimulate cell differentiation, with the additional elevation of Ras activity potentiated by the binding of Crk proteins to the signaling complex resulting in the necessary promotion of cell shape changes. In support of this idea, we note that the loss of Crk proteins did not completely squelch Ras-MAPK signaling, allowing cell differentiation to proceed in *Crk* mutant lenses. Therefore, in response to heightened FGF signaling, Crk proteins act as additional boosters to Ras activity that results in the specific promotion of cell shape changes.

## Materials and methods

### Key resources table

| Reagent type (species) or resource | Designation | Source or reference | Identifiers | Additional information |
|---|---|---|---|---|
| Genetic reagent (*M. musculus*) | *Crk^flox*, *CrkL^flox* | PMID: 19074029 | RRID:MGI:3830069 | Dr. Tom Curran (The Children's Research Institute, Children's Mercy Kansas City) |
| Genetic reagent (*M. musculus*) | *Fgfr3^flox* | PMID: 20582225 | RRID:MGI:4459834 | Dr. Xin Sun (University of California San Diego) |

*Continued on next page*

*Continued*

| Reagent type (species) or resource | Designation | Source or reference | Identifiers | Additional information |
|---|---|---|---|---|
| Genetic reagent (*M. musculus*) | $Frs2\alpha^{flox}$ | PMID: 17868091 | RRID:MGI:3768915 | Dr. Feng Wang (Texas A and M) |
| Genetic reagent (*M. musculus*) | $Rap1a^{flox}$, $Rap1b^{flox}$ | PMID: 18305243 | RRID:MGI:3777607 | Alexei Morozov (National Institutes of Health) |
| Genetic reagent (*M. musculus*) | $Shp2^{flox}$ | PMID: 15520383 | RRID:MGI:3522138 | Gen-Sheng Feng (UCSD) |
| Genetic reagent (*M. musculus*) | $Fgf3^{OVE391}$ | PMID: 7539358 | | Dr. Michael Robinson (Miami University) |
| Genetic reagent (*M. musculus*) | $Fgf3^{OVE393A}$ | PMID: 9640329 | | Dr. Michael Robinson (Miami University) |
| Genetic reagent (*M. musculus*) | $Fgfr1^{\Delta Frs}$ | PMID: 16421190 | RRID:MGI:3620075 | Dr. Raj Ladher (RIKEN Kobe Institute-Center for Developmental Biology) |
| Genetic reagent (*M. musculus*) | $Fgfr1^{Crk}$ | PMID: 26341559 | RRID:MGI:5882534 | Dr. Philipo Soriano (Washington University Medical School) |
| Genetic reagent (*M. musculus*) | $Fgfr2^{flox}$ | PMID: 12756187 | RRID:MGI:3044690 | Dr. David Ornitz (Washington University Medical School) |
| Genetic reagent (*M. musculus*) | $Fgfr4^{-/-}$ | PMID: 9716527 | RRID:MGI:3653043 | Dr. Chu-Xia Deng (National Institute of Health) |
| Genetic reagent (*M. musculus*) | $Grb2^{flox}$ | PMID: 21427701 | RRID:MGI:4949890 | Dr. Lars Nitschke (University of Erlangen-Nürnberg) |
| Genetic reagent (*M. musculus*) | $Pax6^{Le}$-*Cre* (*Le-Cre*) | PMID: 11069887 | RRID:MGI:3045795 | Dr. Ruth Ashery-Padan (Tel Aviv University) |
| Genetic reagent (*M. musculus*) | $Rac1^{flox}$ | PMID: 12759446 | RRID:MGI:2663672 | Dr. Feng-Chun Yang (Indiana University School of Medicine) |
| Genetic reagent (*M. musculus*) | $Rac2^{-/-}$ | PMID: 10072071 | RRID:MGI:3840460 | Dr. Feng-Chun Yang (Indiana University School of Medicine) |
| Genetic reagent (*M. musculus*) | $Kras^{LSL-G12D}$ | PMID: 15093544 | RRID:MGI:3044567 | Mouse Models of Human Cancers Consortium Repository at National Cancer Institute |
| Genetic reagent (*M. musculus*) | $Fgfr1^{flox}$ | Jackson Laboratory | Stock #: 007671 RRID:MGI:3713779 | PMID:16421190 |
| Genetic reagent (*M. musculus*) | $R26\text{-}Rac1^{LSL-G12V}$ | Jackson Laboratory | Stock #: 012361 RRID:MGI:4430563 | PMID:19879843 |
| Genetic reagent (*M. musculus*) | $ROSA^{mTmG}$ | Jackson Laboratory | Stock #: 007676 RRID:MGI:3722405 | PMID:17868096 |
| Cell line (*M. musculus*) | NIH-3T3 | American Type Culture Collection | Cat# CRL-1658, RRID:CVCL_0594 | |
| Transfected construct (synthesized) | TAP-CrkL | PMID: 26527617 | | David J. Glass |
| Antibody | Rabbit anti-C-maf | Santa Cruz Biotechnology | Cat. #: sc-7866 RRID: AB_638562 | IHC (1:200) |
| Antibody | Rabbit anti-CrkL | Santa Cruz Biotechnology | Cat. #: sc-319 RRID: AB_631320 | IHC (1:100), WB (1:1000) |
| Antibody | Mouse anti-CrkL | Santa Cruz Biotechnology | Cat. #: sc-365471 | WB (1:1000) |
| Antibody | Mouse anti-E-cadherin | Sigma | Cat. #: U3254 | IHC (1:200) |
| Antibody | Rabbit anti-Frs2 | Santa Cruz Biotechnology | Cat. #: sc-8318 RRID: AB_2106228 | WB (1:1000) |
| Antibody | Rabbit anti-Grb2 | Santa Cruz Biotechnology | Cat. #: sc-255 RRID: AB_631602 | WB (1:1000) |

*Continued on next page*

*Continued*

| Reagent type (species) or resource | Designation | Source or reference | Identifiers | Additional information |
| --- | --- | --- | --- | --- |
| Antibody | Rabbit anti-Jag1 | Santa Cruz Biotechnology | Cat. #: sc-6011 RRID: AB_649689 | IHC (1:100) |
| Antibody | Mouse anti-Ki-67 | BD Pharmingen | Cat. #: 550609 RRID: AB_393778 | IHC (1:200) |
| Antibody | Rabbit anti-p57 | Abcam | Cat. #: ab75974 | IHC (1:2000) |
| Antibody | Rabbit anti-Pax6 | Covance | Cat. #: PRB-278P RRID: AB_291612 | IHC (1:500) |
| Antibody | Rabbit anti-pCrk (Tyr221) | Cell Signaling | Cat. #: 3491 | IHC (1:200), WB (1:1000) |
| Antibody | Rabbit anti-pCrkL (Tyr207) | Cell Signaling | Cat. #: 3181 | WB (1:1000) |
| Antibody | Rabbit anti-pERK1/2 | Cell Signaling | Cat. #: 4370 | IHC (1:200), WB (1:1000) |
| Antibody | Mouse anti-pERK1/2 | Santa Cruz Biotechnology | Cat. #: sc-7383 | WB (1:1000) |
| Antibody | Rabbit anti-Prox1 | Covance | Cat. No.: PRB-238C | IHC (1:1000) |
| Antibody | Mouse anti-Rac1 | BD Transduction Laboratory | Cat. #: 610652 RRID: AB_397979 | IHC (1:200) |
| Antibody | Rabbit anti-Shp2 | Santa Cruz Biotechnology | Cat. #: sc-280 RRID: AB_632401 | IHC (1:100), WB (1:1000) |
| Antibody | Rabbit anti-α-crystallin | Sam Zigler (National Eye Institute) | | IHC (1:5000) |
| Antibody | Mouse anti-α-SMA | Sigma | Cat. #: C6198 | IHC (1:1000) |
| Antibody | Mouse anti-β-catenin | Sigma | Cat. #: 6F9 | IHC (1:200) |
| Antibody | Rabbit anti-β-crystallin | Sam Zigler (National Eye Institute) | | IHC (1:5000) |
| Antibody | Rabbit anti-γ-crystallin | Sam Zigler (National Eye Institute) | | IHC (1:5000) |
| Recombinant DNA reagent | Ad5CMVCre-eGFP | Gene Transfer Vector Core, University of Iowa, IA | VVC-U of Iowa-1174 | |
| Recombinant DNA reagent | Ad5CMVeGFP | Gene Transfer Vector Core, University of Iowa, IA | VVC-U of Iowa-4 | |
| Peptide, recombinant protein | recombinant human FGF2 | ScienCell | 104–02 | |
| Peptide, recombinant protein | recombinant murine FGF2 | ScienCell | 124–02 | |
| Commercial assay or kit | In situ cell death detection kit, Fluorescein | Sigma | 11684795910 ROCHE | |
| Commercial assay or kit | streptavidin resin | Agilent | 240207 | |
| Chemical compound, drug | Alexa Fluor 488 Phalloidin | ThermoFisher Scientific | A12379 | 1:50 |
| Chemical compound, drug | heparin sodium | Sigma | H3393 | |

## Mice

Mice carrying $Crk^{flox}$, $Crkl^{flox}$, $Fgfr3^{flox}$, $Frs2\alpha^{flox}$, $Rap1a^{flox}$, $Rap1b^{flox}$ and $Shp2^{flox}$ alleles were bred and genotyped as described (*Lin et al., 2007*; *Pan et al., 2008*; *Park and Curran, 2008*; *Su et al., 2010*; *Zhang et al., 2004*). $Fgf3^{OVE391}$ and $Fgf3^{OVE393A}$ were from Dr. Michael Robinson (Miami University, Oxford, OH), $Fgfr1^{\Delta Frs}$ from Dr. Raj Ladher (RIKEN Kobe Institute-Center for Developmental Biology, Kobe, Japan), $Fgfr1^{Crk}$ from Dr. Philipo Soriano (Washington University Medical School, St Louis, MO), $Fgfr2^{flox}$ from Dr. David Ornitz (Washington University Medical School, St Louis, MO), $Fgfr3^{flox}$ from Dr. Xin Sun (University of California San Diego, La Jolla, CA), $Fgfr4^{-/-}$ from Dr. Chu-Xia Deng (National Institute of Health, Bethesda, MD), $Grb2^{flox}$ from Dr. Lars Nitschke (University of Erlangen-Nürnberg, Erlangen, Germany), $Pax6^{Le}$-Cre (*Le-Cre*) from Drs. Ruth Ashery-Padan (Tel Aviv

University, Tel Aviv, Israel) and Richard Lang (Children's Hospital Research Foundation, Cincinnati, OH), $Rac1^{flox}$ and $Rac2^{-/-}$ from Dr. Feng-Chun Yang (Indiana University School of Medicine, Indianapolis, IN) (Ackermann et al., 2011; Ashery-Padan et al., 2000; Brewer et al., 2015; Glogauer et al., 2003; Hoch and Soriano, 2006; Roberts et al., 1999; Robinson et al., 1998; Weinstein et al., 1998; Yu et al., 2003). $Kras^{LSL-G12D}$ mice were obtained from the Mouse Models of Human Cancers Consortium (MMHCC) Repository at National Cancer Institute (Tuveson et al., 2004). $Fgfr1^{flox}$ (Stock No: 007671), $R26-Rac1^{LSL-G12V}$ (Stock No: 012361) and $ROSA^{mTmG}$ (Stock No: 007676) mice were obtained from Jackson Laboratory (Hoch and Soriano, 2006; Muzumdar et al., 2007; Srinivasan et al., 2009). In all conditional knockout experiments, mice were maintained on a mixed genetic background and at least three animals were analyzed for each genotype (Supplementary file 1). We did not observe phenotypic variation in lens development among $Pax6^{Le}$-Cre and $Pax6^{Le}$-Cre;$Crk^{floxl+}$;$Crkl^{flox/+}$ mice, and thus $Pax6^{Le}$-Cre only mice were used as controls. Mouse maintenance and experimentation was performed according to protocols approved by Columbia University Institutional Animal Care and Use Committee.

## Histology, Immunohistochemistry, Immunocytochemistry

Mouse embryos were fixed with 4% paraformaldehyde (PFA) in PBS overnight and paraffin or cryo embedded. The paraffin sections (10 µm) were rehydrated and stained with hematoxylin and eosin (H and E) for histological analysis. Lens sizes were measured as previously described (Cai et al., 2011; Pan et al., 2010). TUNEL staining and immunostaining were performed on the cryosections (8 µm) as previously described (Carbe et al., 2012; Carbe and Zhang, 2011). For phospho-ERK and Shp2 staining, the signal was amplified using a Tyramide Signal Amplification kit (TSA Plus System, PerkinElmer Life Sciences, Waltham, MA). Antibodies used were: anti-Shp2 (Sc-280), anti-C-maf (sc-7866), anti-Crkl (Sc-319), anti-Jag1 (Sc-6011) (all from Santa Cruz Biotechnology, Santa Cruz, CA), anti-pCrkl (Tyr207) (#3181, also recognize pCrk (Tyr221) and anti-pERK1/2 (#4370) (both from Cell Signaling Technology, Boston, MA), anti-P57 (ab75974, from Abcam, Boston, MA), anti-α-SMA (#C6198), anti-β-catenin (6F9), anti-E-cadherin (U3254) (all from Sigma, St. Louis, MO), anti-Ki67 (#550609, BD Pharmingen, San Jose, CA), anti-Crk (#610036), anti-Rac1 (#610651) (both from BD Transduction Laboratory, Franklin Lakes, NJ), anti-Prox1 (PRB-238C) and anti-Pax6 (PRB-278P) (both from Covance, San Diego, CA). Antibodies against α-, β- and γ-crystallins were kindly provided by Sam Zigler (National Eye Institute). Cell proliferation and apoptosis were measured as the ratio of Ki67 or TUNEL-positive cells versus DAPI-positive cells, and analyzed by one-way ANOVA analysis. Alexa Fluor 488 Phalloidin (A12379, ThermoFisher) was used to stain F-actin.

## Lens epithelium explant culture

Postnatal day 0 to day 3 $Crk^{flox/flox}$;$Crkl^{floxflox}$ mice were sacrificed and eyes enucleated. Lenses were then dissected out in lens explant culture medium containing DMEM with 1% BSA (BP1600, Fisher Scientific) and 1:100 dilution of Antibiotic-Antimycotic (15240062, ThermoFisher). Lens capsules were torn open from the posterior before the lens epithelium was peeled off with forceps and pinned down onto a cell culture dish. To delete Crk and Crkl in the lens explants, $2 \times 10^7$ adenoviruses expressing Cre recombinase (Ad5CMVCre-eGFP, Cat #: VVC-U of Iowa-1174, Gene Transfer Vector Core, University of Iowa, IA) were added to the culture of 4 explants for 8 hr one day after explant isolation. GFP-expressing adenoviruses (Ad5CMVeGFP, Cat #: VVC-U of Iowa-4) were used as a control. To induce fiber cell differentiation and elongation, these explants were further cultured in the lens explant culture medium with 2 mg/ml heparin sodium (H3393, Sigma) and 100 ng/ml recombinant murine FGF2 (124–02, ScienCell, Carlsbad, CA ) for 4–5 days.

## Cell protein extract isolation and western blots

$4.24–6.36 \times 10^5$ MEF cells infected with Ad5CMVCre were seeded in 60 mm dishes and serum starved (0.5% FBS in DMEM) for 36–48 hr before being stimulated by 50 ng/ml FGF2 (R and D Systems, Minneapolis, MN) for 5 min at 37°C as previously described (Li et al., 2014). After washing twice in cold PBS, MEF cells were lysed in 160 µl RIPA buffer (50 mM Tris-HCl, pH 8.0, 150 mM NaCl, 1% NP40, 0.5% sodium deoxycholate, 0.1% SDS, 1 µg/ml aprotinin, 1 µg/ml pepstatin, 10 mM sodium pyrophosphate, 1 mM PMSF, 0.2 mM $Na_3VO_4$, 50 mM NaF). Proteins were visualized by infrared-based western blot analysis using an Odyssey SA scanner (LICOR Biosciences, Lincoln, NE).

The signal intensity was quantified using the Odyssey software. The antibodies used were mouse anti-phospho-ERK1/2 (sc-7383, Santa Cruz Biotechnology), anti-phospho-Crk (tyr221) (#3491, Cell Signaling Technology), rabbit-anti-Crkl (sc-319, Santa Cruz biotechnology), mouse-anti-Crkl (sc-365471, Santa Cruz biotechnology), anti-Shp2 (sc-280, Santa Cruz biotechnology), anti-Grb2 (sc-255, Santa Cruz biotechnology), and anti-Frs2 (sc-8318, Santa Cruz biotechnology).

## Immunoprecipitation

NIH3T3 cells from the American Type Culture Collection were tested mycoplasma-free. They were plated at $1 \times 10^6$ cell per 10 cm tissue culture plate and transfected with 2.5 µg TAP-Crkl plasmid (*Hallock et al., 2015*) using lipofectamine 3000 (ThermoFisher, Springfield Township, NJ). 36 hr after the transfection, cells were serum depleted overnight and then were treated with 100 ng/ml recombinant human FGF2 (104–02, ScienCell) for 5 min. Both the treatment group and control were lysed with 1 ml immunoprecipitation buffer (25 mM Tris-HCl, pH 7.4, 1 mM EDTA, 150 mM NaCl, 1% NP-40, 5% glycerol) supplemented with Halt protease inhibitor cocktail (ThermoFisher). Lysates were then incubated with streptavidin resins (240207, Agilent, Santa Clara, CA) to pull down TAP-Crkl, according to manufacturer's recommendations. Resins were later washed in streptavidin binding buffer (10 mM Tris-HCl, pH 7.5, 1 mM EDTA, 150 mM NaCl) twice at 4°C. All pulled-down proteins were eluted in 1X Laemmli SDS sample buffer (1.5% SDS, 9% glycerol, 62.5 mM Tris-HCl, pH 6.8, 0.00025% Bromophenol blue, 2% β-mercaptoethanol). Samples were denatured at 95°C for 5 min before being loaded onto SDS-PAGE gels.

## Acknowledgements

The authors thank Drs. Chu-Xia Deng, Raj Ladher, Richard Lang, Alexei Morozov, Lars Nitschke, David Ornitz, Ruth Ashery-Padan, Philippe Soriano and Feng-Chun Yang for mice. We also thank Drs. Frank Lovicu and Judith West-Mays for advice on lens explant culture. The work was supported by NIH (EY017061 to XZ). The Columbia Ophthalmology Core Facility is supported by NIH Core grant 5P30EY019007 and unrestricted funds from Research to Prevent Blindness (RPB). Xin Zhang is supported by Jules and Doris Stein Research to Prevent Blindness Professorship. Tamica Collins is a recipient of Edwin T Harper Scholarship.

## Additional information

### Funding

| Funder | Grant reference number | Author |
| --- | --- | --- |
| National Eye Institute | EY017061 | Tamica N Collins<br>Yingyu Mao<br>Hongge Li<br>Angela Hong<br>Xin Zhang |
| National Eye Institute | 5P30EY019007 | Xin Zhang |
| Research to Prevent Blindness | Jules and Doris Stein professorship | Xin Zhang |

The funders had no role in study design, data collection and interpretation, or the decision to submit the work for publication.

### Author contributions

Tamica N Collins, Yingyu Mao, Data curation, Formal analysis, Investigation, Methodology, Writing—original draft, Writing—review and editing; Hongge Li, Investigation, Methodology, Writing—review and editing; Michael Bouaziz, Investigation, Writing—review and editing; Angela Hong, Formal analysis, Investigation; Gen-Sheng Feng, Fen Wang, Lawrence A Quilliam, Lin Chen, Taeju Park, Tom Curran, Resources, Writing—review and editing; Xin Zhang, Conceptualization, Resources, Data curation, Formal analysis, Supervision, Funding acquisition, Methodology, Writing—original draft, Project administration, Writing—review and editing

### Author ORCIDs

Xin Zhang  http://orcid.org/0000-0001-5555-0825

### Ethics

Animal experimentation: Mouse maintenance and experimentation was performed according to protocols approved by Columbia University Institutional Animal Care and Use Committee (protocol AAAR0429).

### Decision letter and Author response

Decision letter https://doi.org/10.7554/eLife.32586.018
Author response https://doi.org/10.7554/eLife.32586.019

## Additional files

### Supplementary files

• Supplementary file 1. List of genetic crosses and experimental progenies.
DOI: https://doi.org/10.7554/eLife.32586.015

• Transparent reporting form
DOI: https://doi.org/10.7554/eLife.32586.016

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
