## [Decision Letter]

Thank you for submitting your article "Crk proteins transduce FGF signaling to promote lens fiber cell elongation" for consideration by *eLife*. Your article has been reviewed by two peer reviewers, and the evaluation has been overseen by a Reviewing Editor and Fiona Watt as the Senior Editor. The following individual involved in review of your submission has agreed to reveal his identity: Mark Lewandoski (Reviewer #1).

The reviewers have discussed the reviews with one another and the Reviewing Editor has drafted this decision to help you prepare a revised submission.

Summary:

The reviewers thought that your manuscript utilizes sophisticated genetics and biochemistry to reveal the components and interactions downstream of FGF signaling during the morphogenesis of the mouse lens. They felt that your work has important implications for how molecular signaling controls cell shape. Furthermore, the experiments were described as well executed and the findings as very interesting. The reviewers describe concerns that should be addressed before publication as detailed below.

Essential revisions:

1) There were concerns that "Le-Cre only mice were used as controls." It is important to have Le-Cre in the controls, because Le-Cre itself can have an effect on eye development (PLoS One. 2014 Oct 1;9(10):e109193). But it is also important that littermates are used as controls to rule out differences due to genetic background affecting phenotypes (and the Le-Cre alone effect is background dependent!). This is a big problem as the authors would have to repeat a lot of experiments. Short of doing that work, perhaps the authors can argue that there was little or no variation in phenotypes? Thus the authors should mention how many samples were examined (hopefully at least three) and mention somewhere what variation was observed. It might also help to include a table listing all the genetic crosses, with parental genotypes and the important experimental progeny.

2) Some other histological phenotypes in the Crk/L-cKO mutants are not adequately described. For example, the lens epithelial layer appears missing (Figure 2). Also, there is an abnormal optic cub protrusion posterior to the lens vesicle in Crk/L-cKO mutants. The authors need to describe these phenotypes in more detail, and explain how these phenotypes are affected in other mutant backgrounds. If the lens epithelium is indeed disrupted, could this phenotype affect the maintenance of the polarity of primary fibers and the generation of secondary lens fibers? Also, given the lens-specific deletion of Crk/L, why does the mutation cause the abnormality in the optic cup?

3) The authors emphasize the co-expression of the pCrk/L and pERK in the transitional zone, where proliferation and differentiation of secondary lens fibers occurs, at E14.5. However, the Crk/L-cKO lens phenotype appears between E10.5 and E12.5. The authors should compare the pCrk/L and pER expression at these earlier stages to strengthen their argument.

4). In Figure 3, it appears that the loss of Crk/L mainly affects the substantiation of pERK after FGF2 treatment. Quantification of the western blotting should be very informative here. It would be interesting that Crk/L modify the quantity of FGF-ERK signaling by prolonging ERK activation.

5) "Despite the severe morphological defects, Prox1, Pax6 and multiple forms of crystallin (α,β,γ) were expressed without delay (Figure 2)." The provided data do not demonstrate the normal onset of Pax6. The authors need to modify the statement or provide additional data to support the statement. Also, authors should indicate that the Pax6+ lens epithelium is greatly reduced and shifted to the ventral position in the mutant.

6) Figure 6, the authors should provide quantitative data to demonstrate the partial rescue of the Crk/L-cKO phenotype by Rac1-ca.

7) Figure 6 and Figure 7, are post hoc tests used to confirm where the difference occurs between groups after a statistically significant one-way ANOVA test?

8) In the Abstract, the authors claim that Crk proteins are recruited to the Frs2/Sjp2/Grb2 signaling complex upon FGF signaling. The author should tone down that assertion because there is no direct evidence that Grb2 is a part of the complex.

9) In both Introduction and Discussion, many statements referring to published findings lack references. The authors need to revise these sections and add proper citations.

---

## [Author Response]

Essential revisions:1) There were concerns that "Le-Cre only mice were used as controls." It is important to have Le-Cre in the controls, because Le-Cre itself can have an effect on eye development (PLoS One. 2014 Oct 1;9(10):e109193). But it is also important that littermates are used as controls to rule out differences due to genetic background affecting phenotypes (and the Le-Cre alone effect is background dependent!). This is a big problem as the authors would have to repeat a lot of experiments. Short of doing that work, perhaps the authors can argue that there was little or no variation in phenotypes? Thus the authors should mention how many samples were examined (hopefully at least three) and mention somewhere what variation was observed. It might also help to include a table listing all the genetic crosses, with parental genotypes and the important experimental progeny.

We are aware that Le-Cre mice appear normal in the original FVB/N background but present ocular abnormalities in the CBA/Ca background as shown in the citation above, thus we always use Le-Cre mice as controls instead of wild type (+/+) animals. As described in the revised Materials and methods section, we believe such controls are justified because we did not observe any phenotypic differences between Le-Cre and Le-Cre;Crkflox/+;CrkLflox/+ mice in our mixed genetic background. Moreover, lens development was consistent and reproducible among these controls as shown by the modest variations in our quantifications (see Figure 2 and Figure 4, for example). The revised Materials and methods section now includes a table of all the genetic crosses and confirms that at least three animals are examined for each genotype.

2) Some other histological phenotypes in the Crk/L-cKO mutants are not adequately described. For example, the lens epithelial layer appears missing (Figure 2). Also, there is an abnormal optic cub protrusion posterior to the lens vesicle in Crk/L-cKO mutants. The authors need to describe these phenotypes in more detail, and explain how these phenotypes are affected in other mutant backgrounds. If the lens epithelium is indeed disrupted, could this phenotype affect the maintenance of the polarity of primary fibers and the generation of secondary lens fibers? Also, given the lens-specific deletion of Crk/L, why does the mutation cause the abnormality in the optic cup?

We highlight Figure 2 to show that the lens epithelial layer is diminished but not missing in Crk/L-cKO mutants (Figure 2, arrowheads). Using apical marker Zo-1 and basal marker β1 integrin, we now show that the polarity of the lens fiber cells is also maintained (Figure 2). We also cited earlier work that show the disruption of lens development can lead to disorganization of the retina, which we believe explains the protrusion/abnormality of the optic cup in Crk/L-cKO mutants.

3) The authors emphasize the co-expression of the pCrk/L and pERK in the transitional zone, where proliferation and differentiation of secondary lens fibers occurs, at E14.5. However, the Crk/L-cKO lens phenotype appears between E10.5 and E12.5. The authors should compare the pCrk/L and pER expression at these earlier stages to strengthen their argument.

The pCrk/L and pERK expressions at E12.5 are now added to Figure 1 and Figure 3, respectively.

4). In Figure 3, it appears that the loss of Crk/L mainly affects the substantiation of pERK after FGF2 treatment. Quantification of the western blotting should be very informative here. It would be interesting that Crk/L modify the quantity of FGF-ERK signaling by prolonging ERK activation.

Good advice. We have quantified the western blots in Figure 3, which indeed showed that depletion of Crk/L reduced both the intensity and duration of FGF-ERK signaling. This interesting observation is now described in the text.

5) "Despite the severe morphological defects, Prox1, Pax6 and multiple forms of crystallin (α,β,γ) were expressed without delay (Figure 2)." The provided data do not demonstrate the normal onset of Pax6. The authors need to modify the statement or provide additional data to support the statement. Also, authors should indicate that the Pax6+ lens epithelium is greatly reduced and shifted to the ventral position in the mutant.

We agree. This sentence is revised to only state that these lens markers are still expressed in the Crk/L-cKO mutant lens and it is noted that the lens epithelium is diminished and rotated ventrally.

6) Figure 6, the authors should provide quantitative data to demonstrate the partial rescue of the Crk/L-cKO phenotype by Rac1-ca.

The quantification is presented in Figure 6 (the last two columns).

7) Figure 6 and Figure 7, are post hoc tests used to confirm where the difference occurs between groups after a statistically significant one-way ANOVA test?

As now described in the figure legends, Tukey's multiple comparisons test was used after one-way ANOVA test.

8) In the Abstract, the authors claim that Crk proteins are recruited to the Frs2/Sjp2/Grb2 signaling complex upon FGF signaling. The author should tone down that assertion because there is no direct evidence that Grb2 is a part of the complex.

Agree. The modified sentence now only note that Crk proteins interact with Grb2 upon FGF signaling.

9) In both Introduction and Discussion, many statements referring to published findings lack references. The authors need to revise these sections and add proper citations.

We apologize for the oversight and added additional references as we have originally intended to.